# Status and Determinants of Symptoms of Anxiety and Depression among Food Delivery Drivers in Shanghai, China

**DOI:** 10.3390/ijerph192013189

**Published:** 2022-10-13

**Authors:** Yuxun Peng, Yuqing Shao, Ziyun Li, Ruian Cai, Xiaochen Bo, Chen Qian, Qiao Chu, Jiang Chen, Jianwei Shi

**Affiliations:** 1School of Public Health, Shanghai Jiaotong University School of Medicine, Shanghai 200025, China; 2Department of Sports Medicine, Shanghai Jiao Tong University Affiliated Sixth People’s Hospital, Shanghai 200233, China; 3Shanghai Baoshan District Yanghang Town Community Health Service Center, Shanghai 201901, China; 4Department of General Practice, Yangpu Hospital, Tongji University School of Medicine, Shanghai 200090, China; 5Department of Social Medicine and Health Management, School of Public Health, Shanghai Jiaotong University School of Medicine, Shanghai 200025, China

**Keywords:** work-related factors, anxiety, depression, influencing factor

## Abstract

(1) Background: The psychological status of employees, especially vulnerable populations, has received considerable research attention. However, as a newly emerging and popular occupation in the gig industry, food delivery drivers have received little attention. The majority of these workers are immigrants who are already in a precarious position due to a lack of available jobs, inadequate medical care, poor diets, and communication and acculturation difficulties even before they take these jobs, which involve long working hours and exposure to the elements. (2) Methods: To examine the anxiety and depression symptoms of these workers and possible influencing factors, a cross-sectional study was conducted with a sample of food delivery drivers working for the Meituan Company (one of the largest e-platform companies in China). Anxiety and depression scales were adapted from the GAD-7, and the PHQ-9 was used to assess participants’ related symptoms. Differences were compared in terms of sociodemographic, work situation, and lifestyle variables. Binary logistic regressions were conducted to analyze the effects of various factors on the two psychological dimensions. (3) Results: Among the 657 participants, the proportions of participants reporting anxiety and depression symptoms were 46.0% and 18.4%, respectively. Lack of communication with leaders (OR_AN_ = 2.620, 95% CI: 1.528–4.493, *p* < 0.001; OR_DE_ = 1.928, 95% CI: 1.039–3.577, *p* = 0.037) and poor sleep quality (OR_AN_ = 2.152, 95% CI: 1.587–2.917, *p* < 0.001; OR_DE_ = 2.420, 95% CI: 1.672–3.504, *p* < 0.001) were significant risk factors for both anxiety and depression symptoms. Women (OR = 2.679, 95% CI: 1.621–4.427, *p* < 0.001), those who climbed ≥31 floors per day (OR = 2.415, 95% CI: 1.189–4.905, *p* = 0.015), and those with a high frequency of breakfast consumption (OR = 3.821, 95% CI: 1.284–11.369, *p* = 0.016) were more likely to have anxiety symptoms. Participants who earned less than 5000 RMB (OR = 0.438, 95% CI: 0.204–0.940, *p* = 0.034), were unwilling to seek medical help (OR = 3.549, 95% CI: 1.846–6.821, *p* < 0.001), or had a high frequency of smoking (OR = 5.107, 95% CI: 1.187–21.981, *p* = 0.029) were more likely to be depressive. (4) Conclusion: The existence of communication channels with leaders and good sleep quality are protective factors for anxiety and depression symptoms. Participants who were female, climbed ≥31floors per day, and had a high frequency of eating breakfast were more likely to have anxiety symptoms, while earning less, unwillingness to seek medical help, and a high frequency of smoking were risk factors for depression symptoms.

## 1. Introduction

Adverse psychological health conditions exist in all professions, with common conditions being stress, depression, and anxiety. Up to 41% of the general population will suffer from depression in their lifetime [1]. The lifetime prevalence of anxiety disorders is 3.8–33.7%, and, given comorbidities and primary diagnoses, anxiety disorders constitute three-quarters of all psychiatric disorders [2]. Existing studies have shown that a high level of burnout could have detrimental effects on behavioral and organizational outcomes [3]. Regarding the possible reasons, researchers have found that organizational injustice and restructuring, long working hours, and work–life imbalance are potential causes of depression among working people. For instance, in studies by Tokuyama et al. (2003) [4] and Wieclaw et al. (2006) [5], aspects of the psychosocial work environment, such as effort–reward imbalance, organizational injustice, and undesirable work events, were associated with an increased risk of depression or depressive symptoms. Lorant’s study (2003) identified low socioeconomic status as a risk factor for major depression [6]. Studies have shown an increase in depression in obese individuals [7,8]. One study concluded that a diet that emphasizes fruits can reduce the risk of depression [9]. In addition, Marianna et al. (2012) found strong evidence that age (early or mid-adulthood), female sex, binge drinking, smoking, low socioeconomic status, and negative stressful life events were predictors of depression [10,11]. Fried et al. (2014) found that women were more likely to have physical symptoms such as fatigue, sleep problems, and appetite problems, while men were more likely to have suicidal thoughts [12,13]. Nathaniel et al. (2015) found that poor sleep quality may increase both anxiety and depression symptoms [14]. In Virtanen’s study (2011), long working hours increased the risk of various adverse outcomes, including psychological distress and symptoms of depression and anxiety [15].

Among the various occupations, food delivery driving represents a novel vocation in the gig economy, especially in developed and developing countries with high e-platform development. Existing studies have focused on delivery persons and Uber drivers and found that the characteristics of food delivery drivers have some similarities but differ in that they are mostly part-time and may be mentally healthier than traditional workers because they have greater decision-making capabilities, enabling them to overcome difficulties and enjoy more activities of their choosing; they also have lower psychological strain and higher confidence and self-worth [16,17]. However, other reports have shown that they have worse general mental health and lower job satisfaction than permanent employees [18,19].

These workers are also set apart from more typical delivery staff by other characteristics; most of them are migrants who eat irregularly [20], get less sleep, have inferior health insurance, and are less likely to have a career direction [21], according to Jin et al.’s study (2020). Similarly, Dütsch (2011) noted that because food delivery drivers mostly work on a temporary basis, they lack career prospects, which increases their work stress and is conducive to anxiety [19]. Furthermore, when delivering food, they mostly use electric bikes, which are open to the elements and leave them susceptible to the weather and a poor working environment [22]. They have the added pressure of being forced to deliver food quickly, and, if they are late, they may be fined.

To our knowledge, there is no quantitative analysis of food delivery drivers’ mental health and its influencing factors. To understand the current psychological status of food delivery drivers, this study aims to investigate the occupational mental health and depression of food delivery drivers and to identify the possible influencing factors. We hope to improve the psychological status of vulnerable food delivery drivers in China.

## 2. Materials and Methods

### 2.1. Data Source

In this study, subjects were enrolled from the Meituan Company in Shanghai. Meituan offers services on an e-commerce platform, with takeaway service as its primary offering. Currently, Meituan has a dominant position in the market for online food and beverage takeaway. In 2020, the e-platform had nearly three million food delivery drivers, and, in the fourth quarter of 2021, the peak order volume of Meituan takeaway exceeded 50 million orders per day [23]. After this study was explained to human resources personnel, participants were randomly chosen by their job numbers. The inclusion criteria were as follows: (1) relevant work experience (over a month on the job) as food delivery drivers, and (2) voluntary participation in the survey. An online questionnaire was designed, and the survey was conducted via mobile devices from July through August 2021. A total of 783 questionnaires were distributed; 657 were valid, for a valid response rate of 83.9%.

### 2.2. Measures

The data collection consisted of a self-report survey completed by food delivery drivers that covered sociodemographic variables, work situation, lifestyle, and psychological status.

#### 2.2.1. Outcome Variables

The outcome variables of psychological status (anxiety and depression) in this study were designed on the basis of the Generalized Anxiety Disorder 7-Item Scale (GAD-7) [24] and Patient Health Questionnaire-9 (PHQ-9) [25], respectively. The GAD-7 is a self-rating measure used to assess general anxiety disorder, with a score range of 0–21 [24,26]. The PHQ-9 is the most widely used instrument for screening depression in primary healthcare [27,28], with a total score of 27. For both scales, a score of 10 or more is considered a positive cutoff point for screening, with high sensitivity and specificity [24,28,29,30].

To better target the participants, we specified the GAD-7 regulations and fine-tuned the expression of the PHQ-9. Outcomes were presented on a scale with 10 items representing anxiety and nine items representing depression, each of which was scored on a five-point Likert scale (1 = “never”, 2 = “seldom”, 3 = “sometimes”, 4 = “often”, 5 = “always”). Two numbers equal to 10 points on the original scales were taken as cutoffs (24 for anxiety and 17 for depression, converted in equal proportions).

The final KMO coefficients for the anxiety and depression dimensions were 0.954 and 0.958, respectively, and the significance probability values for Bartlett’s sphericity test were both <0.001, indicating that the scale had high construct validity and was suitable for factor analysis. The AVE values for the psychological dimensions of anxiety and depression were 0.764 and 0.826, respectively, and the CR values were 0.970 and 0.977, respectively, indicating high convergent validity. The square root of the AVE for the two was 0.874 and 0.909, respectively, both greater than the maximum value of the absolute value of the inter-factor correlation coefficient of 0.507, indicating good discriminant validity. Additionally, the Guttman split-half coefficients were 0.943 and 0.944, respectively, indicating good reliability of the two scales.

#### 2.2.2. Independent Variables

The independent variables were as follows: (1) demographics, including gender (male, female), age (≤29, 30–39, and ≥40 years), household registration (Shanghai and other), education (junior high school or less, senior high school, junior college, and undergraduate or more), monthly household income (<5000, 5000–9999, and ≥10,000 RMB), BMI (<18.5, [18.5, 24.0), [24.0, 28.0), ≥28.0 kg/m^2^), medical insurance (Shanghai urban medical insurance, other cities’ urban medical insurance, new rural cooperative medical insurance, commercial medical insurance or other, and no insurance), marital status (unmarried, married, and divorced/widowed), living with family members (yes and no), and willingness to seek medical care (yes and no); (2) work situation, including work type (part-time and full-time), work years (<5 and ≥5 years), daily working hours (<8, 8–10, and ≥11 h), communicating with leaders (yes and no), floors climbed per day (≤10, 11–20, 21–30, and ≥31 floors), transportation (bike, electric bike/motorbike, and van/car/other), and food delivering distance (≤25, 25–50, 51–75, and ≥76 km/day); (3) lifestyle, including daily sleeping time (≤6, 7–8, and ≥9 h), sleep quality (continuous variable, with higher scores indicating poorer sleep quality), regularity of meals (never, seldom, sometimes, often, and always), frequency of breakfast (never, seldom, sometimes, often, and always), eating fruit (never, 1–2, 3–4, 5–6, and ≥7 times/week), smoking (never, seldom, sometimes, often, and always), and drinking (never, seldom, sometimes, often, and always).

### 2.3. Statistical Analysis

SPSS 26.0 software was used in this study. Descriptive statistical analysis methods and chi-square tests were used to analyze whether there were differences among the various influencing factors. Binary logistic regression analysis was performed to screen factors associated with anxiety and depression symptoms and calculate ORs (odds ratios) and their 95% CIs (confidence intervals). All analyses were considered statistically significant at *p* < 0.05 (two-tailed).

## 3. Results

### 3.1. Basic Information

As shown in Table 1, among the 657 participants, most were male (*n* = 467, 71.1%), younger than 30 years old (*n* = 475, 72.3%), unmarried (*n* = 439, 66.8%), and with a household registration outside Shanghai (*n* = 459, 69.9%).

A higher proportion of women than men were aged 40 years or older (15.3%, *p* < 0.001), and a higher proportion of women than men had a BMI below 18.5 (19.5%, *p* < 0.001). The proportion of men who were unmarried was significantly higher than that of women (71.7%, *p* < 0.001). The proportions of men working full time (57.0%, *p* = 0.004) and more than 11 h a day (16.5%, *p* = 0.002) were significantly higher than those of women. Regarding the work situation, a higher proportion of men than women climbed 31 floors or more per day (18.8%, *p* < 0.001) and delivered 76 km or more (21.2%, *p* = 0.003). In terms of lifestyle, a higher proportion of women slept less than or equal to 6 h (38.9%, *p* = 0.027). The proportion of men who never ate regular meals (18.0%, *p* < 0.001) and never ate breakfast (17.3%, *p* = 0.007) was higher, and the proportion of women who never smoked (54.7%, *p* < 0.001) or drank (40.0%, *p* = 0.013) was higher.

### 3.2. Anxiety and Depression Symptoms among Food Delivery Drivers with Different Characteristics

Table 2 summarizes the rates of anxiety and depressive symptoms among food delivery drivers. The proportions of participants reporting anxiety and depression symptoms were 46.0% and 18.4%, respectively. Significantly higher proportions of anxiety symptoms were found in females (58.9%, *p* < 0.001) and among participants who did not live with family members (53.8%, *p* = 0.002), were not willing to seek medical care (61.7%, *p* = 0.001), worked fewer than 5 years (48.3%, *p* = 0.024), had no communication channels with leaders (66.9%, *p* < 0.001), climbed ≥31 floors per day (58.4%, *p* = 0.002), sometimes used electric bikes or motorbikes as transportation tools (49.9%, *p* < 0.001), delivered food 25–50 miles per day (54.4%, *p* < 0.001), slept six or fewer hours per day (52.3%, *p* = 0.016), had poor sleep (*p* < 0.001), sometimes had regular meals (61.2%, *p* < 0.001), sometimes had breakfast (65.5%, *p* < 0.001), ate fruit 1–2 times per week (56.3%, *p* < 0.001), sometimes smoked (62.3%, *p* < 0.001), and often drank (62.1%, *p* < 0.001).

For depression, a higher proportion of participants were unwilling to seek medical care (37.2%, *p* < 0.001), had no communication channels with leaders (32.3%, *p* < 0.001), slept nine or more hours per day (26.8%, *p* = 0.001), had poor sleep (*p* < 0.001), always had regular meals (23.9%, *p* = 0.002), sometimes had breakfast (32.7%, *p* < 0.001), and always smoked (40.4%, *p* < 0.001) and drank (50.0%, *p* < 0.001).

### 3.3. Logistic Regression Analysis of Factors Influencing the Psychology of Food Delivery Drivers

Table 3 shows the results of the logistic regression analysis. Lack of communication with leaders (OR_AN_ = 2.620, 95% CI: 1.528–4.493, *p* < 0.001; OR_DE_ = 1.928, 95% CI: 1.039–3.577, *p* = 0.037) and poor sleep quality (OR_AN_ = 2.152, 95% CI: 1.587–2.917, *p* < 0.001; OR_DE_ = 2.420, 95% CI: 1.672–3.504, *p* < 0.001) were significant risk factors for both anxiety and depression symptoms.

With regard to the presence of anxiety symptoms, we found that women were more likely to have anxiety symptoms than men (OR = 2.679, 95% CI: 1.621–4.427, *p* < 0.001). Those who had been in the job for 5 years or more were significantly less likely to have anxiety symptoms than others (OR = 0.594, 95% CI: 0.355–0.993, *p* = 0.047). Those who climbed 31 floors or more per day were more likely to have anxiety symptoms than those who climbed ≤10 floors per day (OR = 2.415, 95% CI: 1.189–4.905, *p* = 0.015). In regard to lifestyle, sleeping 7–8 h (OR = 0.534, 95% CI: 0.330–0.863, *p* = 0.010) or 9 h (OR = 0.413, 95% CI: 0.197–0.865, *p* = 0.019) and more per day were protective factors for the development of anxiety symptoms compared to shorter sleep periods. Those who sometimes (OR = 3.061, 95% CI: 1.154–8.122, *p* = 0.025), often (OR = 2.966, 95% CI: 1.125–7.817, *p* = 0.028), and always (OR = 3.821, 95% CI: 1.284–11.369, *p* = 0.016) ate breakfast were more likely to be anxious. Compared to those who never ate fruit, participants who ate fruit 1–2 times a week were more likely to have anxiety symptoms (OR = 2.260, 95% CI: 1.214–4.206, *p* = 0.010). Lastly, participants who drank alcohol, whether seldom (OR = 2.240, 95% CI: 1.184–4.239, *p* = 0.013), sometimes (OR = 4.084, 95% CI: 1.900–8.779, *p* < 0.001), often (OR = 3.803, 95% CI: 1.479–9.780, *p* = 0.006), or always (OR = 6.358, 95% CI: 1.396–28.946, *p* = 0.017), were more likely to be anxious than those who did not drink at all.

Regarding the occurrence of depression symptoms, the results showed that participants with a monthly household income of 10,000 RMB or more were less likely to experience depressive symptoms than those who earned less than 5000 RMB (OR = 0.438, 95% CI: 0.204–0.940, *p* = 0.034). Those who were unwilling to seek medical help were more likely to have depressive symptoms (OR = 3.549, 95% CI: 1.846–6.821, *p* < 0.001). In addition, smoking seldom (OR = 4.792, 95% CI: 1.612–14.243, *p* = 0.005), sometimes (OR = 5.049, 95% CI: 1.824–13.971, *p* = 0.002), or always (OR = 5.107, 95% CI: 1.187–21.981, *p* = 0.029) was a risk factor for the development of depression symptoms, with the risk level increasing with frequency.

## 4. Discussion

This study updates our understanding of the current anxiety and depression status of urban food delivery drivers in Shanghai. Very few studies have been conducted on this vulnerable population. The subjects of this study were mainly young and middle-aged men, mostly migrant workers who had been working in Shanghai for a short period of time and were performing this work part-time. The results showed that 46% of the participants reported anxiety symptoms and 18% reported depression symptoms. The prevalence of anxiety was much higher than that reported in previous studies [31,32]. Similar studies on Uber employees, who fill a role similar to that of food delivery drivers, have found the former group to be in a better state of psychological health [33]. This difference may be because Uber employees are not like Chinese food delivery drivers, who have a larger workload, poor transportation, and intense delivery time limits. In addition, different social contexts in different countries at the time of the COVID-19 pandemic may have led to different outcomes. Further comparative studies are needed to determine the exact differences.

In this study, females were more likely to have anxiety symptoms than males. Possible reasons may be that, with intense work and severe overwork, women who are less physically capable may be more stressed than men. Additionally, a study by Pereira-Morales et al. (2019) found that the association between perceived stress and anxiety symptoms is more evident in women [34]. Recent evidence specific to the gig economy finds a growing inequality due to the sex gap in earnings favoring men [35,36], which may further add to the anxiety of female food delivery drivers. Additionally, among our participants, a higher proportion of women than men were unwilling to seek medical care, which would also increase the probability of depression symptoms.

In terms of the work situation, our results showed that excessive daily floor climbing was a risk factor for anxiety symptoms, while longer working years and higher wages were protective factors for anxiety and depression symptoms, respectively. Consistent with Sanne’s view [37], physically harder work and lower income may lead to increased anxiety and depressive conditions. Conversely, longer years of service may help workers work more lightly, thereby reducing career anxiety [38] and perhaps even boosting earnings. However, more research is needed to distinguish the source of the emotional stressor and the occupational role to improve the psychological state of workers [39].

We found that the presence of effective communication channels with the upper echelons of the company was significantly associated with lower rates of both anxiety and depression symptoms. However, 19.8% of our respondents clearly indicated a lack of upward communication. Possible reasons were as follows: (1) while food delivery drivers, as a new form of employment, are distinguished by their flexible working style due to the weakening of subordinate labor relations, they also have the side-effects of weakened organizational relations and a lack of collective consciousness due to their autonomous work habits, unlike traditional workers who have stable work and close organizational relations [20]; (2) in line with Jin’s study (2020) [20], our study found that only 31.1% of subjects had attended university; they had a low level of education overall, and 69.9% were migrant workers who were less socially adjusted than local residents, which may lead to ineffective use of the possible communication channels. Companies should consider the actual situation of these young migrant workers to open up simple and efficient communication channels for them. It is necessary for leaders to value food delivery drivers’ feedback while also guiding them to affirm their value and speak up for themselves.

Consistent with previous studies [40,41], sleep quality influenced both the anxiety and the depression dimensions, and people who experienced anxiety and depression were also more likely to lose sleep [14]. However, in this study, the variable of daily sleeping time was related only to anxiety, probably because the mere length of sleep does not determine the state of life as long as habits are formed. However, the guidelines recommend 7–9 h of sleep per day for healthy adults, and insufficient sleep can certainly have health consequences [42]. Xiao’s study (2019) in China pointed to higher levels of burnout among delivery workers [43] due to severe overwork and lack of rest; consistent with the results of our study, this may be an important risk factor for developing anxiety [10,43,44]. To address this problem, the current takeaway industry should limit working hours [38].

We found that having the habit of eating breakfast and eating fruit 1–2 times a week were risk factors for anxiety symptoms, contradicting the findings of previous studies [45,46]. In addition, our findings support the conclusions that smoking and alcohol abuse are risk factors for anxiety and depression symptoms [47,48]. However, the associations between alcohol consumption and depression symptoms and between smoking and anxiety symptoms were not statistically significant, which may have been due to the characteristics of food delivery drivers or our sample. More research needs to be conducted to explore these issues. However, there is a need to improve the working conditions of food delivery drivers so that they have more room to change their lifestyles.

Several limitations present opportunities for potential research. First, the current cross-sectional questionnaire survey was conducted in one setting, Shanghai. This may limit the generalizability to other geographic regions. Second, respondents may have overestimated or underestimated their degree of anxiety and depression because of the subjective investigation. Third, although the questionnaire controlled for many covariates, the possibility of some residual confounding caused by unmeasured factors should be included, such as a test for perceived stress.

## 5. Conclusions

This study sought to explore anxiety and depression indicators among food delivery drivers. Quantitative analysis was conducted to analyze the results and influencing factors. The subjects of this study were mainly young and middle-aged men; 46% of the participants reported anxiety symptoms, and 18% reported depression symptoms. Female respondents were more likely to have anxiety symptoms than men because of the intense work and inequality in the market. The existence of communication channels with leaders was a protective factor for anxiety and depressive symptoms. Daily sleeping time was related to anxiety, and smoking and alcohol abuse were risk factors for anxiety and depression symptoms. Society should improve the respect for and understanding of service providers and enhance their occupational status and social security.

## Figures and Tables

**Table 1 ijerph-19-13189-t001:** Descriptive analysis of participants.

Item	Classification	*N* (%)			*p*-Value
		Total	Male	Female	
*Demographics*					
Age (year)	≤29	475 (72.3)	353 (75.6)	122 (64.2)	<0.001
	30–39	124 (18.9)	85 (18.2)	39 (20.5)	
	≥40	58 (8.8)	29 (6.2)	29 (15.3)	
Household registration	Shanghai	198 (30.1)	142 (30.4)	56 (29.5)	0.813
	Other	459 (69.9)	325 (69.6)	134 (70.5)	
Education	Junior high school or less	140 (21.3)	100 (21.4)	40 (21.1)	0.150
	Senior high school	154 (23.4)	120 (25.7)	34 (17.9)	
	Junior college	159 (24.2)	110 (23.6)	49 (25.8)	
	Undergraduate or more	204 (31.1)	137 (29.3)	67 (35.3)	
Monthly household income (RMB *)	<5000	152 (23.1)	92 (19.7)	60 (31.6)	0.005
	5000–9999	249 (37.9)	191 (40.9)	58 (30.5)	
	≥10,000	187 (28.5)	137 (29.3)	50 (26.3)	
	Unknown	69 (10.5)	47 (10.1)	22 (11.6)	
BMI (kg/m^2^)	<18.5	65 (9.9)	28 (6.0)	37 (19.5)	<0.001
	[18.5, 24.0)	373 (56.8)	264 (56.5)	109 (57.4)	
	[24.0, 28.0)	153 (23.3)	122 (26.1)	31 (16.3)	
	≥28.0	66 (10.0)	53 (11.3)	13 (6.8)	
Medical insurance	Shanghai urban medical insurance	197 (30.0)	143 (30.6)	54 (28.4)	0.371
	Other cities’ urban medical insurance	188 (28.6)	125 (26.8)	63 (33.2)	
	New rural cooperative medical insurance	92 (14.0)	65 (13.9)	27 (14.2)	
	Commercial medical insurance or other	109 (16.6)	78 (16.7)	31 (16.3)	
	No insurance	71 (10.8)	56 (12.0)	15 (7.9)	
Marital status	Unmarried	439 (66.8)	335 (71.7)	104 (54.7)	<0.001
	Married	165 (25.1)	98 (21.0)	67 (35.3)	
	Divorced/widowed	53 (8.1)	34 (7.3)	19 (10.0)	
Whether living with family members	Yes	417 (63.5)	287 (61.5)	130 (68.4)	0.093
	No	240 (36.5)	180 (38.5)	60 (31.6)	
Willing to seek medical care	Yes	563 (85.7)	405 (86.7)	158 (83.2)	0.237
	No	94 (14.3)	62 (13.3)	32 (16.8)	
*Work situation*					
Work type	Part-time	397 (60.4)	266 (57.0)	131 (68.9)	0.004
	Full-time	260 (39.6)	201 (43.0)	59 (31.1)	
Work years (years)	<5	509 (77.5)	353 (75.6)	156 (82.1)	0.070
	≥5	148 (22.5)	114 (24.4)	34 (17.9)	
Daily working hours (h)	<8	302 (46.0)	197 (42.2)	105 (55.3)	0.002
	8–10	263 (40.0)	193 (41.3)	70 (36.8)	
	≥11	92 (14.0)	77 (16.5)	15 (7.9)	
Communication channels with leaders	Yes	527(80.2)	376 (80.5)	151 (79.5)	0.762
	No	130 (19.8)	91 (19.5)	39 (20.5)	
Floors climbed per day (floor)	≤10	260 (39.6)	169 (36.2)	91 (47.9)	<0.001
	11–20	193 (29.4)	131 (28.1)	62 (32.6)	
	21–30	103 (15.7)	79 (16.9)	24 (12.6)	
	≥31	101 (15.4)	88 (18.8)	13 (6.8)	
Transportation	Bike	72 (11.0)	59 (12.6)	13 (6.8)	0.079
	Electric bike/motorbike	539 (82.0)	374 (80.1)	165 (86.8)	
	Van/car/other	46 (7.0)	34 (7.3)	12 (6.3)	
Food delivery distance (km/day)	≤25	168 (25.6)	119 (25.5)	49 (25.8)	0.003
	25–50	215 (32.7)	136 (29.1)	79 (41.6)	
	51–75	153 (23.3)	113 (24.2)	40 (21.1)	
	≥76	121 (18.4)	99 (21.2)	22 (11.6)	
*Lifestyle*					
Daily sleeping time (h)	≤6	218 (33.2)	144 (30.8)	74 (38.9)	0.027
	7–8	357 (54.3)	256 (54.8)	101 (53.2)	
	≥9	82 (12.5)	67 (14.3)	15 (7.9)	
Regularity of meals	Never	101 (15.4)	84 (18.0)	17 (8.9)	<0.001
	Seldom	113 (17.2)	87 (18.6)	26 (13.7)	
	Sometimes	227 (34.6)	155 (33.2)	72 (37.9)	
	Often	128 (19.5)	75 (16.1)	53 (27.9)	
	Always	88 (13.4)	66 (14.1)	22 (11.6)	
Frequency of breakfast	Never	97 (14.8)	81 (17.3)	16 (8.4)	0.007
	Seldom	155 (23.6)	113 (24.2)	42 (22.1)	
	Sometimes	168 (25.6)	121 (25.9)	47 (24.7)	
	Often	135 (20.5)	83 (17.8)	52 (27.4)	
	Always	102 (15.5)	69 (14.8)	33 (17.4)	
Frequency of eating fruit (times/week)	Never	122 (18.6)	91 (19.5)	31 (16.3)	0.097
	1–2	293 (44.6)	219 (46.9)	74 (38.9)	
	3–4	162 (24.7)	103 (22.1)	59 (31.1)	
	5–6	46 (7.0)	30 (6.4)	16 (8.4)	
	≥7	34 (5.2)	24 (5.1)	10 (5.3)	
Frequency of smoking	Never	273 (41.6)	169 (36.2)	104 (54.7)	<0.001
	Seldom	114 (17.4)	91 (19.5)	23 (12.1)	
	Sometimes	138 (21.0)	105 (22.5)	33 (17.4)	
	Often	80 (12.2)	58 (12.4)	22 (11.6)	
	Always	52 (7.9)	44 (9.4)	8 (4.2)	
Frequency of drinking	Never	208 (31.7)	132 (28.3)	76 (40.0)	0.013
	Seldom	169 (25.7)	129 (27.6)	40 (21.1)	
	Sometimes	176 (26.8)	130 (27.8)	46 (24.2)	
	Often	66 (10.0)	44 (9.4)	22 (11.6)	
	Always	38 (5.8)	32 (6.9)	6 (3.2)	

* RMB: Chinese yuan. The average exchange rate between USD and CNY from July to August 2021 was 6.481.

**Table 2 ijerph-19-13189-t002:** Description of anxiety and depressive symptoms among food delivery drivers.

Item	Classification	Anxiety Symptoms			Depression Symptoms		
		No *n* (%)	Yes *n* (%)	*p*-Value	No *n* (%)	Yes *n* (%)	*p*-Value
Total		355 (54.0)	302 (46.0)		536 (81.6)	121(18.4)	
** *Demographics* **							
Sex	Male	277 (59.3)	190 (40.7)	<0.001	379 (81.2)	88 (18.8)	0.658
	Female	78 (41.1)	112 (58.9)		157 (82.6)	33 (17.4)	
Age (year)	≤29	259 (54.5)	216 (45.5)	0.207	377 (79.4)	98 (20.6)	0.052
	30–39	60 (48.4)	64 (51.6)		107 (86.3)	17 (13.7)	
	≥40	36 (62.1)	22 (37.9)		52 (89.7)	6 (10.3)	
Household registration	Shanghai	113 (57.1)	85 (42.9)	0.305	157 (79.3)	41 (20.7)	0.320
	Other	242 (52.7)	217 (47.3)		379 (82.6)	80 (17.4)	
Education	Junior high school or less	79 (56.4)	61 (43.6)	0.771	111 (79.3)	29 (20.7)	0.493
	Senior high school	79 (51.3)	75 (48.7)		130 (84.4)	24 (15.6)	
	Junior college	89 (56.0)	70 (44.0)		133 (83.6)	26 (16.4)	
	Undergraduate or more	108 (52.9)	96 (47.1)		162 (79.4)	42 (20.6)	
Monthly household income (RMB)	<5000	81 (53.3)	71 (46.7)	0.542	118 (77.6)	34 (22.4)	0.388
	5000–9999	127 (51.0)	122 (49.0)		202 (81.1)	47 (18.9)	
	≥10,000	107 (57.2)	80 (42.8)		157 (84.0)	30 (16.0)	
	Unknown	40 (58.0)	29 (42.0)		59 (85.5)	10 (14.5)	
BMI (kg/m^2^)	<18.5	35 (53.8)	30 (46.2)	0.583	51 (78.5)	14 (21.5)	0.685
	[18.5, 24.0)	198 (53.1)	175 (46.9)		310 (83.1)	63 (16.9)	
	[24.0, 28.0)	81 (52.9)	72 (47.1)		123 (80.4)	30 (19.6)	
	≥28.0	41 (62.1)	25 (37.9)		52 (78.8)	14 (21.2)	
Medical insurance	Shanghai urban medical insurance	115 (58.4)	82 (41.6)	0.517	158 (80.2)	39 (19.8)	0.289
	Other cities’ urban medical insurance	103 (54.8)	85 (45.2)		159 (84.6)	29 (15.4)	
	New rural cooperative medical insurance	47 (51.1)	45 (48.9)		76 (82.6)	16 (17.4)	
	Commercial medical insurance or other	56 (51.4)	53 (48.6)		91 (83.5)	18 (16.5)	
	No insurance	34 (47.9)	37 (52.1)		52 (73.2)	19 (26.8)	
Marital status	Unmarried	244 (55.6)	195 (44.4)	0.431	353 (80.4)	86 (19.6)	0.326
	Married	82 (49.7)	83 (50.3)		141 (85.5)	24 (14.5)	
	Divorced/widowed	29 (54.7)	24 (45.3)		42 (79.2)	11 (20.8)	
Whether living with family members	Yes	244 (58.5)	173 (41.5)	0.002	348 (83.5)	69 (16.5)	0.103
	No	111 (46.3)	129 (53.8)		188 (78.3)	52 (21.7)	
Willing to seek medical care	Yes	319 (56.7)	244 (43.3)	0.001	477 (84.7)	86 (15.3)	<0.001
	No	36 (38.3)	58 (61.7)		59 (62.8)	35 (37.2)	
** *Work situation* **							
Work type	Part-time	214 (53.9)	183 (46.1)	0.935	322 (81.1)	75 (18.9)	0.698
	Full-time	141 (54.2)	119 (45.8)		214 (82.3)	46 (17.7)	
Work years (years)	<5	263 (51.7)	246 (48.3)	0.024	423 (83.1)	86 (16.9)	0.062
	≥5	92 (62.2)	56 (37.8)		113 (76.4)	35 (23.6)	
Daily working hours (h)	<8	177 (58.6)	125 (41.4)	0.091	243 (80.5)	59 (19.5)	0.521
	8–10	133 (50.6)	130 (49.4)		220 (83.7)	43 (16.3)	
	≥11	45 (48.9)	47 (51.1)		73 (79.3)	19 (20.7)	
Communication channels with leaders	Yes	312 (59.2)	215 (40.8)	<0.001	448 (85.0)	79 (15.0)	<0.001
	No	43 (33.1)	87 (66.9)		88 (67.7)	42 (32.3)	
Floors climbed per day (floor)	<10	161 (61.9)	99 (38.1)	0.002	216 (83.1)	44 (16.9)	0.852
	11–20	104 (53.9)	89 (46.1)		157 (81.4)	36 (18.7)	
	21–30	48 (46.6)	55 (53.4)		82 (79.6)	21 (20.4)	
	≥31	42 (41.6)	59 (58.4)		81 (80.2)	20 (19.8)	
Transportation tool	Bike	56 (77.8)	16 (22.2)	<0.001	55 (76.4)	17 (23.6)	0.459
	Electric bike/motorbike	270 (50.1)	269 (49.9)		444 (82.4)	95 (17.6)	
	Van/car/other	29 (63.0)	17 (37.0)		37 (80.4)	9 (19.6)	
Food delivering distance (km/day)	<25	113 (67.3)	55 (32.7)	<0.001	139 (82.7)	29 (17.3)	0.247
	25–50	98 (45.6)	117 (54.4)		180 (83.7)	35 (16.3)	
	51–75	83 (54.2)	70 (45.8)		126 (82.4)	27 (17.6)	
	≥76	61 (50.4)	60 (49.6)		91 (75.2)	30 (24.8)	
** *Lifestyle* **							
Daily sleeping time (h)	≤6	104 (47.7)	114 (52.3)	0.016	166 (76.1)	52 (23.9)	0.001
	7–8	197 (55.2)	160 (44.8)		320 (86.8)	47 (13.2)	
	≥9	54 (65.9)	28 (34.1)		60 (73.2)	22 (26.8)	
Sleep quality				<0.001			<0.001
Regularity of meals	Never	80 (79.2)	21 (20.8)	<0.001	91 (90.1)	10 (9.9)	0.002
	Seldom	77 (68.1)	36 (31.9)		102 (90.3)	11 (9.7)	
	Sometimes	88 (38.8)	139 (61.2)		173 (76.2)	54 (23.8)	
	Often	53 (41.4)	75 (58.6)		103 (80.5)	25 (19.5)	
	Always	57 (64.8)	31 (35.2)		67 (76.1)	21 (23.9)	
Frequency of breakfast	Never	78 (80.4)	19 (19.6)	<0.001	86 (88.7)	11 (11.3)	<0.001
	Seldom	95 (61.3)	60 (38.7)		138 (89.0)	17 (11.0)	
	Sometimes	58 (34.5)	110 (65.5)		113 (67.3)	55 (32.7)	
	Often	61 (45.2)	74 (54.8)		112 (83.0)	23 (17.0)	
	Always	63 (61.8)	39 (38.2)		87 (85.3)	15 (14.7)	
Frequency of eating fruit (times/week)	Never	86 (70.5)	36 (29.5)	<0.001	95 (77.9)	27 (22.1)	0.757
	1–2	128 (43.7)	165 (56.3)		240 (81.9)	53 (18.1)	
	3–4	87 (53.7)	75 (46.3)		136 (84.0)	26 (16.0)	
	5–6	33 (71.7)	13 (28.3)		38 (82.6)	8 (17.4)	
	≥7	21 (61.8)	13 (38.2)		27 (79.4)	7 (20.6)	
Frequency of smoking	Never	163 (59.7)	110 (40.3)	<0.001	257 (94.1)	16 (5.9)	<0.001
	Seldom	76 (66.7)	38 (33.3)		96 (84.2)	18 (15.8)	
	Sometimes	52 (37.7)	86 (62.3)		90 (65.2)	48 (34.8)	
	Often	37 (46.3)	43 (53.8)		62 (77.5)	18 (22.5)	
	Always	27 (51.9)	25 (48.1)		31 (59.6)	21 (40.4)	
Frequency of drinking	Never	142 (68.3)	66 (31.7)	<0.001	195 (93.8)	13 (6.3)	<0.001
	Seldom	102 (60.4)	67 (39.6)		154 (91.1)	15 (8.9)	
	Sometimes	67 (38.1)	109 (61.9)		121 (68.8)	55 (31.3)	
	Often	25 (37.9)	41 (62.1)		47 (71.2)	19 (28.8)	
	Always	19 (50.0)	19 (50.0)		19 (50.0)	19 (50.0)	

**Table 3 ijerph-19-13189-t003:** Logistic regression of factors associated with anxiety and depression symptoms among food delivery drivers.

Item	Classification	Anxiety Symptoms		Depression Symptoms	
		OR (95% CI)	*p*-Value	OR (95% CI)	*p*-Value
** *Demographics* **					
Sex	Male (reference)				
	Female	2.679 (1.621, 4.427)	<0.001	1.175 (0.618, 2.233)	0.623
Monthly household income (RMB)	<5000 (reference)				
	5000–9999	1.063 (0.619, 1.824)	0.825	0.825 (0.417, 1.632)	0.581
	≥10,000	0.747 (0.416, 1.342)	0.329	0.438 (0.204, 0.940)	0.034
	Unknown	0.942 (0.457, 1.941)	0.872	0.833 (0.306, 2.270)	0.721
Willing to seek medical care	Yes (reference)				
	No	1.424 (0.775,2.615)	0.254	3.549 (1.846, 6.821)	<0.001
** *Work situation* **					
Work years (years)	<5 (reference)				
	≥5	0.594 (0.355, 0.993)	0.047	1.256 (0.687, 2.297)	0.460
Communication channels with leaders	Yes (reference)				
	No	2.620 (1.528, 4.493)	<0.001	1.928 (1.039, 3.577)	0.037
Floors climbed per day	≤10 (reference)				
	11–20	1.124 (0.669, 1.887)	0.659	1.039 (0.523, 2.068)	0.912
	21–30	1.232 (0.648, 2.341)	0.525	1.152 (0.499, 2.661)	0.740
	≥31	2.415 (1.189, 4.905)	0.015	0.620 (0.240, 1.601)	0.324
** *Lifestyle* **					
Daily sleeping time (h)	≤6 (reference)				
	7–8	0.534 (0.330, 0.863)	0.010	0.727 (0.396, 1.336)	0.305
	≥9	0.413 (0.197, 0.865)	0.019	1.064 (0.460, 2.460)	0.885
Sleep quality		2.152 (1.587, 2.917)	<0.001	2.420 (1.672, 3.504)	<0.001
Frequency of breakfast	Never (reference)				
	Seldom	1.533 (0.648, 3.625)	0.331	0.660 (0.199, 2.196)	0.498
	Sometimes	3.061 (1.154, 8.122)	0.025	1.351 (0.393, 4.645)	0.633
	Often	2.966 (1.125, 7.817)	0.028	1.106 (0.309, 3.957)	0.877
	Always	3.821 (1.284, 11.369)	0.016	0.232 (0.039, 1.386)	0.109
Frequency of eating fruit (times/week)	Never (reference)				
	1–2	2.260 (1.214, 4.206)	0.010	0.760 (0.361, 1.603)	0.472
	3–4	1.461 (0.743, 2.876)	0.272	0.794 (0.347, 1.819)	0.586
	5–6	0.759 (0.295, 1.955)	0.568	0.888 (0.261, 3.018)	0.849
	≥7	0.619 (0.202, 1.898)	0.402	0.396 (0.107, 1.463)	0.165
Frequency of smoking	Never (reference)				
	Seldom	0.558 (0.274, 1.136)	0.108	4.792 (1.612, 14.243)	0.005
	Sometimes	0.694 (0.322, 1.498)	0.352	5.049 (1.824, 13.971)	0.002
	Often	0.584 (0.261, 1.305)	0.190	2.873 (0.898, 9.190)	0.075
	Always	0.355(0.098, 1.285)	0.114	5.107 (1.187, 21.981)	0.029
Frequency of drinking	Never (reference)				
	Seldom	2.240 (1.184, 4.239)	0.013	0.805 (0.260, 2.495)	0.707
	Sometimes	4.084 (1.900, 8.779)	<0.001	2.759 (0.926, 8.220)	0.068
	Often	3.803 (1.479, 9.780)	0.006	1.894 (0.515, 6.966)	0.336
	Always	6.358 (1.396, 28.946)	0.017	3.516 (0.609, 20.307)	0.160

## Data Availability

The data presented in this study are openly available in [FigShare] at [https://doi.org/10.6084/m9.figshare.21322071.v1].

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
