# Peer review of "Status and Determinants of Symptoms of Anxiety and Depression among Food Delivery Drivers in Shanghai, China"

_ijerph, 2022, doi:10.3390/ijerph192013189_

Round 1
Reviewer 1 Report
The manuscript contains data of interest, especially due to the type of workers that make up the study sample. but the authors must carry out a thorough review of the current version of aspects of both content and form. My main concerns and suggestions are:
- The keywords can be improved (influencing?) –
the manuscript studies symptoms of anxiety and depression, but both in the title and throughout the text (including tables) reference is made to anxiety and depression. This aspect must be carefully reviewed, since a clinical evaluation has not been carried out. but merely screening (presence of symptoms).
– Based on the criterion that the presence of symptomatology in the workers has simply been evaluated, the association analyzes (linear regressions) lose interest and the conclusions derived may not be relevant. That two variables are associated positively or negatively may not mean anything beyond a numerical data that reaches statistical significance. Thus, the anxiety and depression screening scores have a cut-off point of a possible positive case (yes/no) and this is how it should be treated both conceptually and analytically (logistic and non-linear regressions).
- Numerous variables have been evaluated with little precision (i.e. regularity of meals, frequency of breakfast, smoking and drinking). This allows only an approximation, which assumes that significant data will require future studies with a more precise approach that not only evaluates the frequency of consumption but also the amount of tobacco and alcohol consumed. This is also a factor against using linear regressions, since psychometrically most variables are not interval scales.
- The tables can be improved, using only one column for the mean ± SD and together with the contrasts of F/t values, a column must be generated with the calculation of the effect sizes. It is necessary to provide the reliability values in the registered sample for the anxiety and depression scales used.
– I advise the authors to update and expand the bibliography selected in the manuscript, which will allow them to delve into basic and practical aspects related to their results. The paper by Pereira-Morales et al. (2019, Current Psychology, 38, 66-74) introduces aspects of perceived stress in depression and anxiety symptoms that are very useful given the characteristics of the food delivery drivers sample. In addition, the mention of limitations is general and excessively simple, it requires more reflection. A reference to what pandemic situation was being experienced at the time of data recording in July of 2021 would also be of interest.
Reviewer 2 Report
The authors presented interesting results of qualitative study covered food delivery drivers’ mental health and its influencing factors for the new emerging occupation of food delivery drivers. The majority of these workers are immigrants who are already in a precarious position due to lack of available jobs, inadequate medical care, and other negative factors even before they take these jobs, which come with long working hours exposed to the elements. I have some comments:
1) Discuss the potential association anxiety and depression in food delivery drivers with burnout. The aspect of client contacts and the potential negative emotions associated with it are also subject to discussion. For example, see follow studies: DOIs: 10.15275/rusomj.2018.0307, 10.3390/ijerph17217738, etc.
2) Gender is major factor for potential stratification. The differences between men and women were much less pronounced than I expected. This needs to be discussed in great detail, as a parallel analysis of subgroups singled out by gender might be appropriate. Specify, were working conditions the same for men and women?
Round 2
Reviewer 1 Report
The authors have carried out an in-depth review of the manuscript according to my indications, having greatly improved both its content and its form.
It is advisable to take care of some aspects of the citation, such as, for example, in the introduction section, a reference appears with names and year, which is then included together with another one numbered in parentheses.
Reviewer 2 Report
I approve revised version of paper.